

# Soil copper sources and biogeochemical processes under different land uses: insights from stable copper isotopes in the Cambisols of Southwest China

Man Liu[1], Guilin Han[2] and Qian Zhang[3]

[1] Wuhan Botanical Garden, Chinese Academy of Sciences, Wuhan, China
[2] Institute of Earth Sciences, China University of Geosciences (Beijing), Beijing, China
[3] Institute of Geographic Sciences and Natural Resources Research, Chinese Academy of Sciences, Beijing, China

Corresponding author
Guilin Han, hanguilin@cugb.edu.cn

## ABSTRACT

Copper (Cu), a toxic trace element, has extensively accumulated in soils due to intensive human activities. Yet current knowledge on Cu sources and biogeochemical processes in Cambisols under varying land uses is scant. In this study, Cu contents, stable isotope compositions ($\delta^{65}Cu$) and their correlations with soil physicochemical properties in soil profiles across different land uses were analyzed, including cropland, abandoned cropland and orchard, shrub-grass land, and secondary forest in a karst watershed of Southwest China. Soil Cu contents in cropland (mean 44.9 mg kg$^{-1}$) were significantly higher than those in abandoned cropland and orchard (mean 37.7 mg kg$^{-1}$), and much higher than those in natural lands (mean 26.3 mg kg$^{-1}$). In agricultural lands, cropland soils (mean –0.216‰) were significantly $^{65}Cu$-depleted compared to abandoned cropland and orchard (0.020‰), resulting from applying $^{65}Cu$-depleted fungicides. Soil $\delta^{65}Cu$ values in shrub-grass land and secondary forest land exhibited within a wide range, from –0.627‰ to 0.338‰, attributed to Cu isotope fractionations during pedogenic processes. Soil $\delta^{65}Cu$ values decreased with increasing soil depth and were positively correlated with soil organic carbon (SOC) contents, but negatively correlated with the chemical index of alteration (CIA), indicating the influences of clay mineral sorption, organic complexation, and leaching processes on soil $\delta^{65}Cu$ patterns. In different regions, $\delta^{65}Cu$ values in Cambisols increased in sub-humid climate, but decreased in humid climate with increasing annual precipitation. This study underscores the key roles of mineral sorption, organic complexation, and leaching processes in affecting $\delta^{65}Cu$ patterns in Cambisols.

## INTRODUCTION

Copper (Cu), as a potentially toxic trace element, can have detrimental effects on soil biota and fertility when excessively accumulated in soils (*Komarek et al., 2010*). The high concentration of Cu in soils is often attributed to human activities such as mining and smelting, agricultural practices, and urbanization development (*Lai et al., 2024*; *Zheng*

*et al., 2024*). Excessive Cu accumulation not only threatens drinking water and food safety but also ultimately poses a significant risk to human health (*Adams et al., 2024*; *Zeng & Han, 2020*). In soils, Cu primarily exists as the form of Cu(I) and Cu(II), with the latter being more soluble and bio-available (*Grybos et al., 2007*). Dissolved Cu in soils can be interconverted with Cu-containing/adsorbing sulfides, hydroxides, organic matter-mineral complexes, and other insoluble compounds through various chemical and physical processes (*Minkina et al., 2019*), thereby affecting Cu mobility and bio-availability. Thus, predicting the environmental fate of soil Cu necessitates an understanding of its sources and biogeochemical processes (*Kim, Nevitt & Thiele, 2008*).

Land use has the potential to significantly impact the accumulation and bioavailability of Cu in soils (*Prăvălie et al., 2021*). With regards to Cu sources, parent materials and plants are primarily contributor of soil Cu in natural lands (*Fekiacova, Cornu & Pichat, 2015*). In contrast, agricultural land often gains exogenous Cu from farmyard manure and fungicides, and industrial land receives it from Cu-contaminated wastewater and dust, while urban land frequently accumulates Cu from solid waste and traffic exhaust (*Babcsányi et al., 2016*; *Kříbek et al., 2018*; *Nikolaeva et al., 2021*). In terms of Cu biogeochemical processes, land use also plays a crucial role in regulating transformation and translocation processes of Cu. This is accomplished by influencing factors such as soil pH, organic matter abundance, redox condition, microbial activity, leaching, erosion, and plant assimilation (*Liu et al., 2014*; *Maher & Larson, 2007*; *Nguyen et al., 2024*; *Ryan et al., 2014*; *Weinstein et al., 2011*). Currently, few studies have been carried out to investigate Cu sources and biogeochemical processes in relation to different land uses.

In agricultural production, the application of fungicides is a primary way to accumulate Cu in topsoil (*Babcsányi et al., 2016*). Prolonged use of fungicide led to topsoil Cu contents reaching 1,200–1,500 mg kg$^{-1}$ in European vineyards (*Chaignon et al., 2003*), which considerably exceeded the 14 mg kg$^{-1}$ background of European soils (*Lado, Hengl & Reuter, 2008*). In particular, soil carbonates contribute significantly to Cu accumulation through the precipitation of $CuCO_3$ (*Blotevogel et al., 2018*). Generally, excessive Cu negatively impacts soil biota and agroecosystems, such as reducing microbial communities and enzyme activities (*Komarek et al., 2010*). In environments with excessive Cu levels, microbial selection pressure can promote the growth of tolerant microbes, subsequently leading to alterations in microbial community diversity and function (*Fagnano et al., 2020*).

Stable Cu isotope ratios ($^{65}Cu/^{63}Cu$, $\delta^{65}Cu$) are currently employed as a useful tool for elucidating Cu sources and biogeochemical processes (*Fekiacova, Cornu & Pichat, 2015*; *Liu et al., 2014*; *Pappoe et al., 2024*). In relevant environmental research, Cu isotopes offer insights into Cu behaviors in responses to both natural processes and anthropogenic activities (*Wiederhold, 2015*). Natural processes such as chemical weathering, pedogenic processes, and soil-plant interactions significantly influence the migration and transformation of Cu and $\delta^{65}Cu$ patterns in soil profiles (*Bigalke et al., 2010*). These natural processes lead to isotopic fractionation of Cu in soils through mechanisms like redox reactions during chemical weathering (*Zheng et al., 2024*), adsorption by metal (oxyhydr)oxides and clay minerals (*Li, Liu & Li, 2015*), complexation with organic matter (*Ryan et al., 2014*), precipitation in carbonates (*Maher & Larson, 2007*), and plant

assimilation (*Weinstein et al., 2011*). Additionally, anthropogenic activities, particularly mining and smelting, agricultural activities, and urban waste and traffic fumes can significantly alter the Cu isotopic compositions in soils through the introduction of exogenous Cu (*Babcsányi et al., 2016*; *Bigalke, Weyer & Wilcke, 2011*; *Fekiacova, Cornu & Pichat, 2015*; *Kříbek et al., 2018*; *Nikolaeva et al., 2021*). For instance, *Kříbek et al. (2018)* reported that the contaminated soils were notably $^{65}$Cu-enriched (with $\delta^{65}$Cu values from 0.13‰ to 0.76‰) due to the oxidative leaching of Cu minerals, compared to uncontaminated soils (which ranged from −0.17‰ to 0.14‰). Regarding agricultural inputs, *Babcsányi et al. (2016)* examined $\delta^{65}$Cu compositions in fungicides between −0.49‰ and 0.31‰, while *Fekiacova, Cornu & Pichat (2015)* reported the $\delta^{65}$Cu values in farmyard manure (*e.g.*, pig slurry) ranging from 0.12‰ to 0.52‰. Stable isotope analyses of these exogenous Cu-containing materials, including tailing wastewater, fungicides, and farmyard manure, are instrumental in distinguishing between anthropogenic and natural Cu sources, thereby evaluating their environmental fate (*Blotevogel et al., 2018*).

The karst region of Southwest China is characterized by a notably fragile ecosystem (*Han et al., 2020*). Its soils, classified as Cambisols, are marked by slow formation, thin layers, uneven distribution, and a high susceptibility to erosion (*Liu, Han & Zhang, 2020*). These characteristics make them particularly vulnerable to agricultural activities (*Liu, Han & Zhang, 2020*). Restoring the ecosystems after prolonged agricultural cultivation poses significant challenges due to their limited environmental carrying capacities (*Liu et al., 2016*). Additionally, the intensification of agricultural practices has been observed to gradually deplete the nutrient levels in karst soils while simultaneously accumulating potentially toxic Cu (*Liu, Han & Zhang, 2020*; *Zhang et al., 2019*). However, there remains a gap in understanding of the Cu sources and biogeochemical processes in the karst region under varying land uses. The objectives of this study are: (i) to investigate the vertical variations of Cu content and $\delta^{65}$Cu values in soil profiles across different land uses, (ii) to explore soil Cu sources and primary biogeochemical processes, and (iii) to comparatively analyze $\delta^{65}$Cu patterns of Cambisols in different regions.

## MATERIALS AND METHODS

### Study area

The study area is located in the Chenqi watershed, a karst critical zone in Puding County, Southwest China (26°15′16″–26°16′5″N, 105°46′14″–105°46′48″E) (Fig. 1). The Chenqi watershed, covering an area of approximately 1.54 km$^2$, is also a long-term observation zone under the purview of the Puding Karst Ecosystem Research Station, Chinese Academy of Sciences (*Liu, Han & Zhang, 2020*). Th region is characterized by a subtropical monsoon climate, with an average annual temperature of 15 °C and an annual precipitation of 1,315 mm (*Zhang & Han, 2023*). The typical karst landform in the watershed is characterized by a central depression, encircled by hills on three sides, and an average altitude of approximately 1,350 m (*Yue et al., 2020*). The calcareous soils found on the hilltops and mountainside are primarily developed from the Middle Triassic Guanling Formation limestone (*Zhao et al., 2010*). Quaternary sediments located at the foot of the

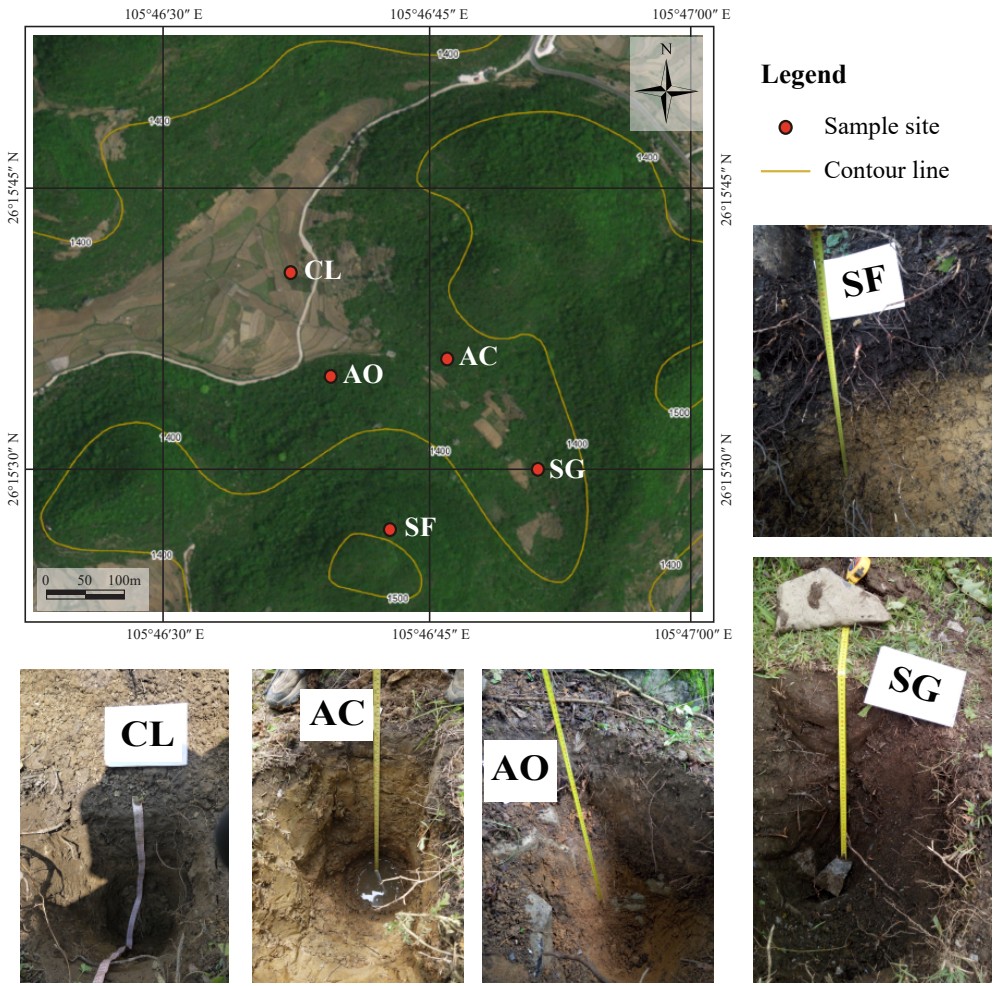

**Figure 1** **Location of sampling sites and soil profile pictures.** Soil profile pictures were cited from *Liu et al. (2019)*.

mountain and in the central depression are largely derived from the translocated materials of surrounding hills (*Green et al., 2019*). The soils, primarily derived from limestone and Quaternary sediments, are predominantly classified as Eutric Cambisols based on the soil classification system approved by the International Union of Soil Sciences (*IUSS Working Group, 2022*). The area of land covered by Eutric Cambisols accounts for more than 90% of the total area of the Chenqi watershed. The distribution of soil layer is discontinuous, ranging from 10 to 160 cm in thickness with an average of 70 cm (*Liu, Han & Zhang, 2020*). Puding County encompasses an area of 336 km$^2$ dedicated to cropland, constituting 31% of the county's land area *Anshun Municipal Bureau of Statistics, 2025*. Over the recent three years, fungicide consumption in agricultural production has averaged 70.15 tons annually, with an application intensity of 208.8 kg km$^{-2}$ (*Anshun Municipal Bureau of Statistics, 2025*).

With population growth and rising food demand, intensive farming practices have led to serious soil degradation over the past 50 years (*Liu et al., 2016*). In response, the Grain for Green Program (GGP) was implemented to restore the ecological function of these degraded lands (*Wang et al., 2017*). As a result, many sloped croplands were abandoned and left to evolve naturally. Consequently, the Chenqi watershed currently encompasses a diverse range of land types, including forest, shrubland, grassland, cropland, and abandoned cropland (Fig. 1).

## Sample collection

Soil sites were selected from the five distinct land use types, based on the level of agricultural disturbance: cropland (CL), abandoned cropland (AC), abandoned orchard (AO), shrub-grass land (SG), and secondary forest (SF) (*Liu et al., 2019*). At each site, three soil profiles were excavated, each separated by a distance of 1 meter. The depth of each soil profile was 70 cm. Given the rapid vertical variations in soil properties in surface soils, soil samples were collected with an interval of 10 cm depth at 0–30 cm layer and an interval of 20 cm depth at 30–70 cm layer. To mitigate the effects of soil heterogeneity, three soil samples from one depth layer within a soil site were evenly mixed to form one sample. All soil samples were passed through a two mm sieve and air-dried at room temperature (25 °C). Dominant vegetation near each soil site was collected and freeze-dried for plant samples. Additionally, bedrock samples were collected from the exposed limestone, removing the regolith of the rock surface. Information regarding land use history, soil profile, and dominant plant species for each soil site is shown in Table 1.

## Sample analyses

The proportions of soil particle were measured using a laser particle size analyzer (Mastersizer 2000; Malvern, Worcestershire, UK), with a precision of ±1%. Soil organic carbon (SOC) contents were determined using a total carbon analyzer, with a precision of ±0.01%. Soil, plant, and bedrock samples were finely ground ($<74\,\mu m$) prior to digestion. A strong acid mixture ($HNO_3$, HCl, HF, and $HClO_4$) was employed for digestion (*Zheng, Han & Liang, 2023*). The concentrations of the major elements (including aluminum (Al), calcium (Ca), sodium (Na), potassium (K), iron (Fe), manganese (MN), titanium (Ti)) in the digested solution were determined using Inductively Coupled Plasma Optical Emission Spectrometry (ICP-OES, Optima 5300DV; PerkinElmer, Springfield, IL, USA). Cu concentration in the digested solution was determined using Inductively Coupled Plasma Mass Spectrometry (ICP-MS, Elan DRC-e; PerkinElmer). To ensure the accuracy of the concentration analyses for all elements, quality control measures, including blank sample and standard sample assessments, were stringently applied. The recovery yields of the digestion procedure for each element exceeded 98.3%, while procedural blank rates remained below 0.3%. Additionally, the relative standard deviation (RSD) for duplicate measurements did not surpass 5%. All element concentrations were measured at the Institute of Geographic Sciences and Natural Resources Research, Chinese Academy of Sciences, China.

**Table 1** Descriptions of land use history and soil profile.

| Soil site | Land use type | Land use history | Dominant plant species | Soil profile |
|---|---|---|---|---|
| CL | Cropland | Long-term plowing; fertilization by chemical fertilizer, and farmyard manure; application of pesticide and fungicide; non-returned crop residues. | *Zea mays* *Brassica napus* | 0–20 cm (A horizon): brawn, block structure, tight, abundant roots and debris; 20–70 cm (B horizon): yellow, block structure, tight, no rootlet. |
| AC | Abandoned cropland | Abandoned cropland for 3 years; dominated by dwarf weeds. | *Conyza canadensis* *Digitaria sanguinalis* *Chrysanthemum indicum* | 0–6 cm (A horizon): brawn, block structure, tight, abundant roots and debris; 6–70 cm (B horizon): yellow, block structure, tight, no rootlet. |
| AO | Abandoned orchard | Abandoned pear orchard for 8 years; with a history of fertilization and application of pesticide and fungicide; dominated by arbors and shrubs. | *Pyrus* *Ailanthus altissima* *Kalopanax septemlobus* | 0–10 cm (A horizon): black humus layer, fine grain, loose, abundant roots and debris; 10–70 cm (B horizon): red, clayey, tight, no rootlet. |
| SG | Shrub-grass land | Dominated by gramineous herbages and shrubs; without agricultural disturbance. | *Eleusine indica* *Rosa cymosa* *Kalopanax septemlobus* | 0–15 cm (A horizon): black humus layer, coarse grain, loose, abundant roots and debris; 15–70 cm (B horizon): red, clayey, tight, no rootlet. |
| SF | Secondary forest | Mixed evergreen and broadleaved deciduous forest; without agricultural disturbance. | *Tilia tuan* *Pinus tabuliformis* *Quercus fabri* | 0–10 cm (A horizon): black humus layer, coarse grain, loose, abundant roots; 10–35 cm (B horizon): brawn, block structure, tight, few roots; 35–70 cm (C horizon): red, block structure, tight, no rootlet. |

## Analyses of stable Cu isotope ratios

The AG MP-1 anion-exchange resin (100–200 mesh, chloride form; Bio-Rad, Hercules, CA, USA) was packed into a 1.6 mL polypropylene column (BioRad, California, USA). Prior to loading, the resin was washed three times with five ml of seven mol $L^{-1}$ HCl, 0.001% $H_2O_2$, and five ml of ultrapure water to equilibrate the resin environment. The digested samples were loaded onto the resin, followed by the addition of seven ml of seven mol $L^{-1}$ HCl and 0.001% $H_2O_2$ to elute impurities. Subsequently, the pure Cu solution was steamed dry and then dissolved in diluted $HNO_3$. The Ni concentration in the sample was tested after separation to ensure the Ni/Cu ratio <0.01, thereby avoiding interference with the stable Cu composition analysis (*Ren et al., 2022*). The anion exchange technology achieved Cu recovery yields exceeding 95%, with procedural blank rates remaining below 0.2%. All separation processes for the Cu solution were conducted at the Nu Surficial Environment & Hydrological Geochemistry Laboratory (Nu-SEHGL) at the China University of Geosciences (Beijing), China.

Stable Cu isotope ratios were measured using a Multi-collector Inductively Coupled Plasma Mass Spectrometer (MC-ICP-MS) instrument (Nu Plasma III; Nu Instrument, Wrexham, United Kingdom), operated in low-resolution mode at the Nu-SEHGL. To mitigate instrument mass bias, standard-sample bracketing (SSB) and pre-calibrated

internal standard solutions were utilized (*Zeng & Han, 2020*). The stable Cu isotope ratios in soil, plant, and bedrock samples were expressed as $\delta^{65}$Cu (‰), as equation following:

$$\delta^{65}Cu = \left[ \frac{(^{65}Cu/^{63}Cu)_{sample}}{(^{65}Cu/^{63}Cu)_{ERM-AE633}} - 1 \right] \times 1000 \qquad (1)$$

where ERM-AE633 was used as the standard material, as opposed to NIST-SRM-976 which has been widely used in the past. Unfortunately, the global supply of NIST-SRM-976 is nearly depleted (*Moeller et al., 2012*), necessitating the switch to ERM-AE633. The measurement sequence incorporated the reference material of the ERM-AE633 (Institute for Reference Materials and Measurements, IRMM, Belgium). The measured $\delta^{65}$Cu values of the NIST-SRM-976 were $-0.01$‰ $\pm 0.05$‰ relative to the ERM-AE633. All $\delta^{65}$Cu data are reported normalized to the NIST-SRM976.

## Enrichment factor (*EF*$_{Cu}$) of Cu in soil

The enrichment factor (*EF*) method was employed to calculate the degree of Cu enrichment in the soils. Titanium (Ti) is an element that exhibits resistance to chemical weathering and is minimally impacted by anthropogenic activities (*Tessier, Campbell & Bisson, 1979*). Thus, Ti is frequently employed as a reference element to indicate the enrichment degree of other heavy metal elements (*Sezgin et al., 2003*; *Yuan et al., 2020*). The formula for the calculation of the enrichment factor is as follows:

$$EF_{Cu} = (Cu/Ti)_{sample}/(Cu/Ti)_{background} \qquad (2)$$

where $(Cu/Ti)_{sample}$ refers to the ratio of measured Cu content to Ti content in soil sample, $(Cu/Ti)_{background}$ represents the average ratio of Cu content to Ti content (mg kg$^{-1}$) in the Guizhou soil of China (*Chen et al., 1991*). The $EF_{Cu} < 1$ suggests the absence of Cu pollution, $1 \leq EF_{Cu} < 2$ indicates slight Cu pollution, $2 \leq EF_{Cu} < 5$ indicates moderate Cu pollution, and $EF_{Cu} \geq 5$ denotes significant Cu pollution in soils (*Barbieri, 2016*).

## Chemical index of alteration (CIA)

The chemical index of alteration was employed to evaluate the magnitude of chemical weathering in weathered soils (*Kurtz et al., 2000*). The formula is presented below:

$$CIA = \left[ Al_2O_3/\left(Al_2O_3 + CaO^* + Na_2O + K_2O\right) \right] \times 100 \qquad (3)$$

where $Al_2O_3$, $Na_2O$, and $K_2O$ (mol kg$^{-1}$) represent their mole fraction in whole soil and $CaO^*$ (mol kg$^{-1}$) indicates the Ca mole fraction in the silicate component. Higher CIA values are indicative of increased chemical weathering in the soil.

## Statistical analyses

A one-way analysis of variance (ANOVA), complemented by Tukey's HSD test, was employed to discern significant differences in soil Cu contents and $\delta^{65}$Cu values across varied land uses. Linear regression analysis was leveraged to examine the relationships between soil $\delta^{65}$Cu values and soil physicochemical properties in soil profiles under different land uses. Stepwise regression analysis was used to explore the primary factor affecting soil Cu contents and $\delta^{65}$Cu value of Cambisols in the karst region. Additionally,

non-linear regression analysis was implemented to explore the relationships between $\delta^{65}Cu$ values and annual precipitation in different regions. All statistical analyses were executed using the SPSS and SigmaPlot software packages.

## RESULTS

### Soil physiochemical properties under different land uses

Clay proportions in soils ranged from 14.5% to 24.6%, with an average of 19.6% (Fig. 2A). Soils in secondary forest land and abandoned orchard presented higher clay particles compared to other lands. In secondary forests, clay proportions increased significantly with increasing soil depth, while those exhibited a slight decrease with depth in other land-use types. Silt proportion in soils ranged from 75% to 86%, showing variations along soil profiles that were inverse to those of clay proportions (Fig. 2B). The highest SOC content was observed in secondary forest soils (over 60 g kg$^{-1}$ in topsoil), whereas in other land-use types, those were below 40 g kg$^{-1}$, with the lowest average content (8.7 g kg$^{-1}$) found in cropland (Fig. 2C). Generally, decrease in SOC content with increasing soil depth was noted across all land uses. The $Al_2O_3$ contents in soils varied between 9.9% and 18.8%, with the highest content in cropland and the lowest content in secondary forest land (Fig. 2D). Besides shrub-grass land, $Al_2O_3$ contents in other lands increased with increasing soil depth. For all land uses except secondary forest, $Fe_2O_3$ contents ranged from 6% to 9% and typically exhibited a slight decrease with increasing soil depth (Fig. 2E). However, in secondary forest soils, $Fe_2O_3$ contents rose significantly from 4.0% to 6.7% with soil depth. MnO contents in shrub-grass land and abandoned cropland and orchard slightly decreased with increasing soil depth, ranging between 0.10% and 0.18% (Fig. 2F).

Secondary forest land soils had the lowest MnO content, which declined from 0.06% to 0.02% with increasing soil depth. In cropland soils, MnO contents decreased from 0.16% to 0.06% within the 0–30 cm layer, but then increased to 0.16% at the profile's bottom. The CIA values for cropland, abandoned cropland and orchard, and shrub-grass land soils demonstrated a slight uptick with increasing soil depth, ranging from 69 to 76 (Fig. 2G). In contrast, secondary forest land exhibited a more pronounced increase in CIA values, rising significantly from 59 to 72 as soil depth increased. With the exception of cropland soils, the $EF_{Cu}$ values for other land soils spanned a range of 0.6 to 1.2 and commonly did not show intensive variation with an increase in soil depth (Fig. 2H). However, the $EF_{Cu}$ values in cropland soils were lower in the top layer (0–20 cm depth) with an average of 1.2, compared to those found below 20 cm depth, which averaged 1.7.

### Soil Cu contents and $\delta^{65}Cu$ values under different land uses

Soil Cu contents in cropland (mean 44.9 mg kg$^{-1}$) were significantly higher than those in abandoned cropland and orchard (mean 37.7 mg kg$^{-1}$), and considerably greater than those in the shrub-grass land and secondary forest land (mean 26.3 mg kg$^{-1}$) (Fig. 3B). Soil Cu contents generally increased with increasing soil depth in cropland and abandoned cropland, while those decrease with soil depth in abandoned orchard (Fig. 3A). Noteworthy is the lack of significant vertical variations in Cu contents under shrub-grass and secondary forest.

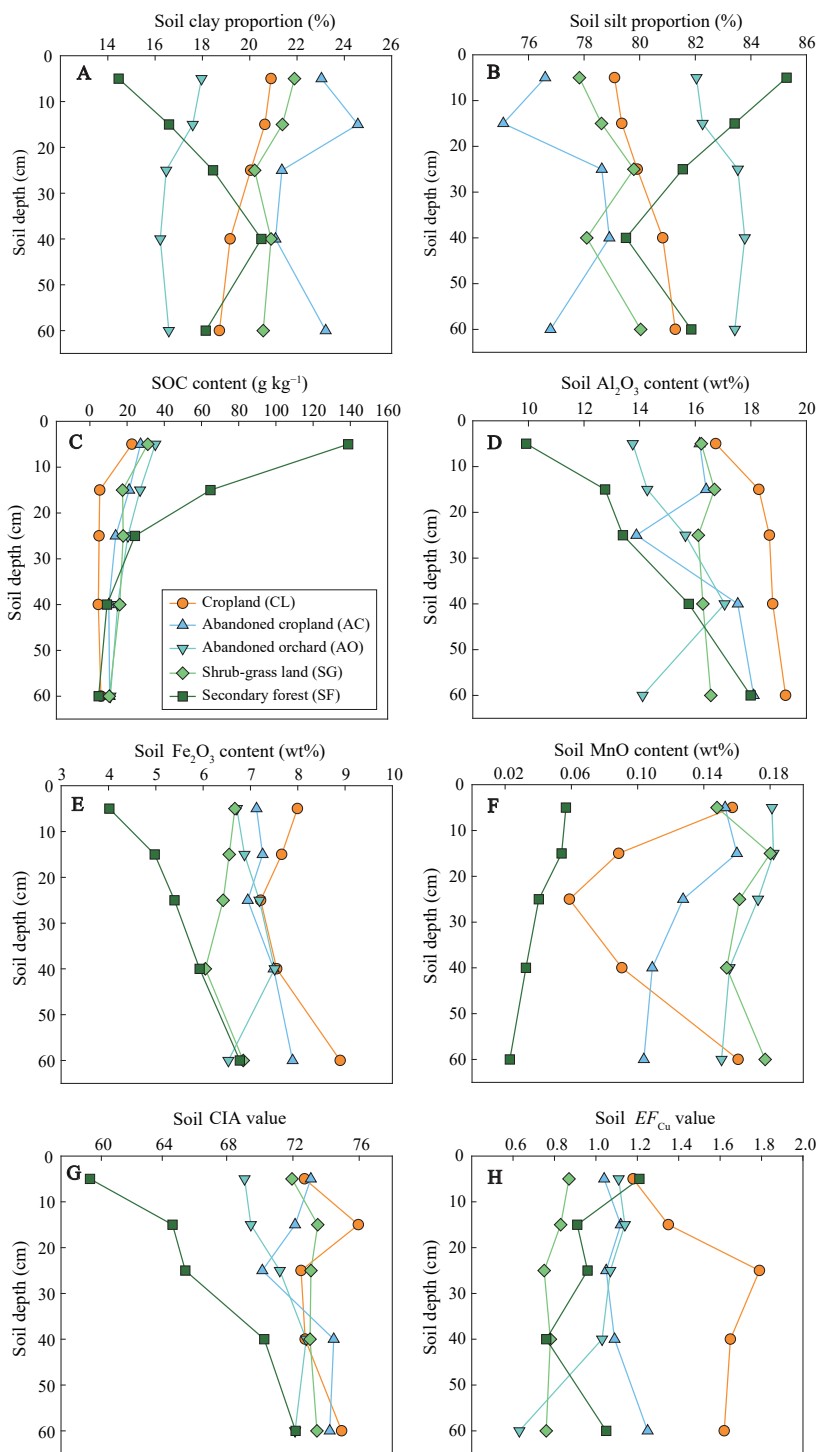

**Figure 2 Soil physicochemical properties in the profiles under different land uses.** Soil physicochemical properties include clay proportion (A), silt proportion (B), SOC content (C), $Al_2O_3$ content (D), $Fe_2O_3$ content (E), MnO content (F), CIA value (G), and $EF_{Cu}$ value (H).

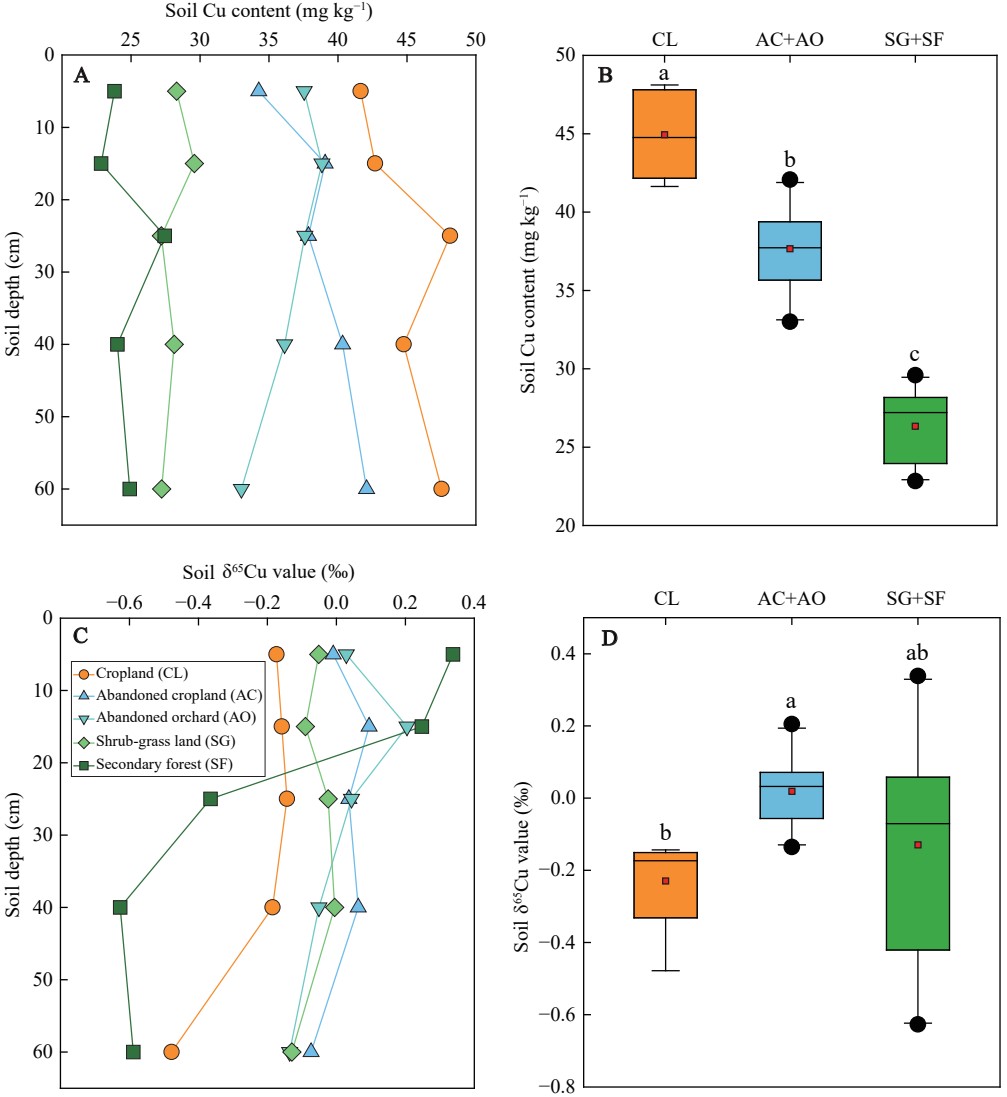

**Figure 3  Soil Cu contents and δ⁶⁵Cu values in the profiles under different land uses.** Different lower-cases in (B) and (D) indicate significant differences in soil Cu contents and $\delta^{65}Cu$ values among the lands with different agricultural disturbances at the level of $p < 0.05$.

Soils in cropland (mean $\delta^{65}Cu$: $-0.216‰$) exhibited a significant depletion in $^{65}Cu$ compared to those in abandoned cropland and orchard (mean $\delta^{65}Cu$: $0.020‰$) (Fig. 3D). In the secondary forest land and shrub-grass land, soil $\delta^{65}Cu$ values ranged from $0.338‰$ to $-0.627‰$, with an average of $-0.130‰$. Notably, the $\delta^{65}Cu$ values in all lands decreased as soil depth increased (Fig. 3C).

## Relationships between soil δ⁶⁵Cu values and soil physicochemical properties

The results of linear regression analysis indicated a significant correlation between soil $\delta^{65}Cu$ values in secondary forest land and soil physicochemical properties at a significance

level of $p < 0.05$ (Fig. 4). Comparatively, the relationships observed in other land use types were weak or non-significant (*i.e.*, $p > 0.05$ and $R^2 < 0.6$). This was mainly attributed to the limited range of $\delta^{65}$Cu values in those profiles, which hindered statistical robustness. Hence, the regression models with a goodness-of-fit ($R^2$) exceeding 0.4 were also trustworthy. Specifically, soil $\delta^{65}$Cu values were negatively correlated with clay proportions in secondary forest land, whereas the correlation was positive in cropland (Fig. 4A). Conversely, soil $\delta^{65}$Cu values showed a positive correlation with silt proportions in secondary forest land but a negative correlation between them in cropland (Fig. 4B). Additionally, soil $\delta^{65}$Cu values were positively correlated with SOC contents in secondary forest land and abandoned orchard (Fig. 4C), while demonstrating a negative correlation with $Al_2O_3$ contents in the same land types (Fig. 4D). Furthermore, soil $\delta^{65}$Cu values were negatively correlated with $Fe_2O_3$ contents across secondary forest land, cropland, and shrub-grass land (Fig. 4E). Positive correlations were observed between soil $\delta^{65}$Cu values and MnO contents in secondary forest land and abandoned orchard but negative correlations between them in cropland and shrub-grass land (Fig. 4F). Finally, soil $\delta^{65}$Cu values were found to have negative correlations with CIA values in secondary forest land and abandoned orchard (Fig. 4G), while no significant correlation was identified with $EF_{Cu}$ values across all lands (Fig. 4H).

The results of the stepwise regression analysis revealed that $Fe_2O_3$ content, $Al_2O_3$ content, and $EF_{Cu}$ value were the principal factors affecting soil Cu content in the karst region (Table 2). Among these, $Fe_2O_3$ content was the most significant predictor of soil Cu content. In comparison, SOC content and MnO content emerged as the primary determinants of the soil $\delta^{65}$Cu value in the region. Notably, SOC content was more important than MnO content for predicting soil $\delta^{65}$Cu value.

# DISCUSSION

## Identification of soil Cu sources under different land uses

Soil Cu primarily originates from parent materials (or bedrock) and external input (*Fekiacova, Cornu & Pichat, 2015*). The Chenqi watershed, a typical agricultural watershed distant from mining and smelting areas and urban centers (*Liu, Han & Zhang, 2020*), relies on limestone, agricultural sources and plant inputs as primary contributors to soil Cu. Soil Cu contents in agricultural lands (including cropland, abandoned cropland and orchard) exceeded the regional background value of 32 mg kg$^{-1}$ (*Chen et al., 1991*). Conversely, natural lands (including secondary forest and shrub-grass land) exhibited significantly lower in soil Cu content compared to this background value (Fig. 3B). Similarly, the $EF_{Cu}$ values were predominantly above 1 in agricultural lands (Fig. 2H), indicating slight Cu pollution in these soils. In comparison, the $EF_{Cu}$ values in the natural lands were below 1, indicating that their primary source from bedrock and plants. Thus, these findings suggested that external inputs from agricultural activities contributed to soil Cu accumulation in cropland, abandoned cropland and orchard.

Soil $\delta^{65}$Cu values did not significantly correlate with $EF_{Cu}$ values across all lands (Fig. 4H), suggesting that soil Cu dynamics were influenced by multiple sources or

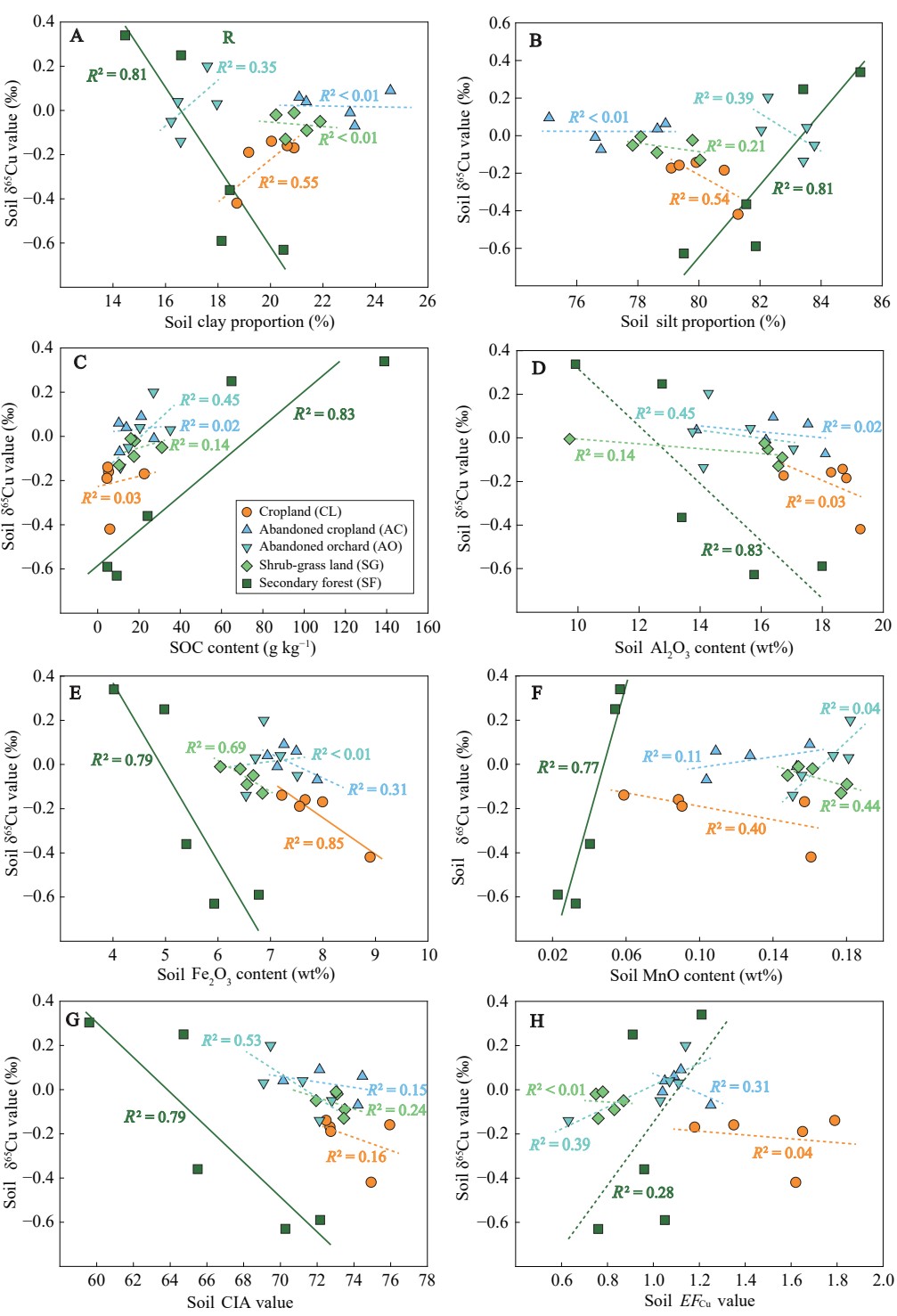

**Figure 4 Relationships between soil physicochemical properties and $\delta^{65}Cu$ values under different land uses.** Soil physicochemical properties include clay proportion (A), silt proportion (B), SOC content (C), $Al_2O_3$ content (D), $Fe_2O_3$ content (E), MnO content (F), CIA value (G), and $EF_{Cu}$ value (H). The solid and dotted fitted lines represent the significance levels of $p < 0.05$ and $p > 0.05$, respectively.

**Table 2  Results of stepwise regression analysis.**

| Dependent variable | Explanatory variable | Unstandardized coefficient | | Standardized coefficient | $t$ | $p$ | VIF | $R^2$ | Adjust $R^2$ | $F$ |
|---|---|---|---|---|---|---|---|---|---|---|
| | | $B$ | Standard error | $Beta$ | | | | | | |
| Soil Cu content | Constant | −5.130 | 4.038 | | −1.271 | 0.218 | | | | |
| | $Fe_2O_3$ content | 6.578 | 0.948 | 0.856 | 6.939 | <0.001 | 3.188 | 0.9 | 0.886 | 62.922 $p < 0.001$ |
| | $EF_{Cu}$ value | 14.247 | 2.109 | 0.530 | 6.754 | <0.001 | 1.293 | | | |
| | $Al_2O_3$ content | −1.281 | 0.430 | −0.360 | −2.980 | 0.007 | 3.056 | | | |
| Soil $\delta^{65}$Cu value | Constant | −0.479 | 0.090 | | −5.291 | <0.001 | | 0.576 | 0.538 | 14.958 $p < 0.001$ |
| | SOC content | 0.006 | 0.001 | 0.677 | 4.797 | <0.001 | 1.035 | | | |
| | MnO content | 2.134 | 0.615 | 0.490 | 3.469 | 0.002 | 1.035 | | | |

processes. To identify the potential soil Cu sources, $\delta^{65}$Cu values in plant and bedrock samples near soil sites were analyzed, using them as endmembers (Fig. 5). The $\delta^{65}$Cu values of limestone ranged from −0.637‰ to −0.582‰, those closely align with the $\delta^{65}$Cu composition (mean −0.608‰) in the C horizon of soil profile under secondary forest (Fig. 5A). This observation corroborates previous findings that $\delta^{65}$Cu values in the C horizon often resemble those in bedrock (*Zheng et al., 2024*). For the plant samples, there were significant differences in $\delta^{65}$Cu values between grass (aboveground tissues) and arbor (leaf), with ranges of −0.243‰ to −0.115‰ and −0.863‰ to −0.386‰, respectively (Fig. 5B). Globally, trees exhibit significantly depleted $\delta^{65}$Cu values (−1.8‰ to −0.8‰) compared to grasses (−0.2‰ to 0.7‰) (*Zheng et al., 2024*). Furthermore, $\delta^{65}$Cu compositions from agricultural sources, primarily fungicides (−0.49‰ to 0.31‰, *Babcsányi et al., 2016*) and farmyard manure (0.12‰ to 0.52‰, *Fekiacova, Cornu & Pichat, 2015*), were included as additional endmembers (Fig. 5A).

The mixing of multiple sources significantly influenced soil $\delta^{65}$Cu compositions. To discern the origins of soil Cu, the scatter plot of 1/[Cu] (*i.e.,* reciprocal transformations of soil Cu contents) *versus* $\delta^{65}$Cu for soil samples and other potential sources was extensively employed (*Ren et al., 2022*; *Zheng, Han & Liang, 2023*). In the agricultural lands, the soil samples were distributed between the agricultural source endmember and the bedrock endmember (Fig. 5A). Notably, soil samples from cropland were more proximate to the $^{65}$Cu-depleted fungicides compared to those from abandoned cropland and orchard. This alignment with practical observations can be attributed to the cessation of fungicide application post-abandonment. Crop litter input barely contributed to soil Cu due to the non-return of straw during the harvest season in this area (*Liu, Han & Zhang, 2020*). Additionally, $^{65}$Cu-depleted litter with its decomposition typically leads to diminished $\delta^{65}$Cu value in the topsoil (*Weinstein et al., 2011*). Nevertheless, soil $\delta^{65}$Cu values remained relatively consistent at depths of 0–30 cm (Fig. 3C), corroborating this hypothesis. In comparison, plant sources showed some contribution to soil Cu in the abandoned cropland and orchard, due to vegetation restoration and litter return. Therefore, the topsoil (0–10 cm layer) exhibited slightly $^{65}$Cu-depletion relative to the deeper soil at the 10–20 cm depth in both lands (Fig. 3C). Yet, $\delta^{65}$Cu values in topsoil (mean 0.015‰) markedly deviated from the plant endmember (mean −0.625‰) (Fig. 5), suggesting that plants contributed

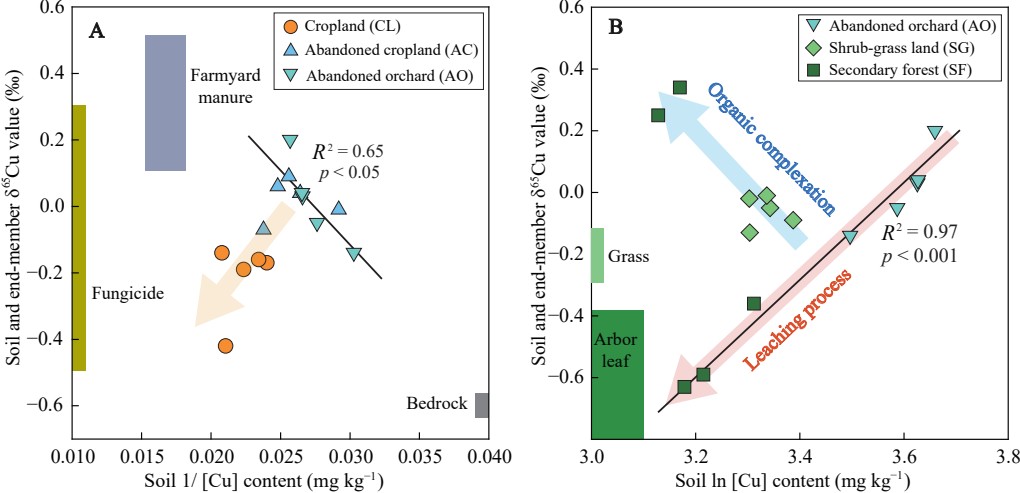

**Figure 5** **Relationships between soil δ⁶⁵Cu values and Cu contents to discriminate the possible sources and processes of soil Cu.** The variables 1/[Cu] and ln[Cu] represent the reciprocal and logarithmic transformations of soil Cu contents, respectively. Endmembers for bedrock and plants determined through the analysis of local samples. Endmembers for farmyard manures and fungicides were cited from *Fekiacova, Cornu & Pichat (2015)* and *Babcsányi et al. (2016)*, respectively.

minimally to soil Cu in agricultural lands. In the natural lands, the primary sources of soil Cu were unequivocally plants and bedrock. However, both plant and bedrock endmembers displayed ⁶⁵Cu-depletion relative to soil samples, resulting in that soil samples did not distribute between plant and bedrock endmembers (Fig. 5B). This result implied that stale Cu isotope fractionation *via* biogeochemical processes affected soil δ⁶⁵Cu composition.

## Biogeochemical processes of Cu under different land uses

Soil δ⁶⁵Cu compositions are also affected by various biogeochemical processes, encompassing redox reactions, mineral dissolution and precipitation, mineral sorption, organic complexation, and plant uptake (*Liu et al., 2014*; *Maher & Larson, 2007*; *Ryan et al., 2014*; *Weinstein et al., 2011*). The distribution patterns of δ⁶⁵Cu in soil profiles and their correlations with soil physiochemical properties can indicate the key biogeochemical processes affecting Cu in specific environment (*Babcsányi et al., 2016*; *Fekiacova, Cornu & Pichat, 2015*). Dissolved Cu(II) generally enriches the heavier isotope compared to Cu(I)-sulfide minerals during redox reactions (*Kimball et al., 2009*). Therefore, soil δ⁶⁵Cu values tend to rise significantly with increasing soil depth in the land with frequent redox transitions (*Zheng et al., 2024*), because ⁶⁵Cu-enriched Cu(II) moves downwards under leaching process. However, the δ⁶⁵Cu values in the profiles of all lands diminished with increasing soil depth (Fig. 3C). Moreover, the Cambisols in the karst region have not undergone frequent redox transitions during their pedogenic processes. Thus, the impact of redox reactions on soil δ⁶⁵Cu composition were not dominant.

The reaction procedure of mineral dissolution is strongly influenced by soil pH and organic ligands (*Murphy, Oelkers & Lichtner, 1989*). Acidic environments facilitate the

dissolution of Cu-containing minerals such as sulfides, carbonates, and hydroxide minerals, resulting in a preferential release of $^{65}$Cu-enriched Cu(II) into the soil solution (*Fekiacova, Cornu & Pichat, 2015*). However, Cambisols developed from limestone are weakly alkaline (*Liu, Han & Zhang, 2020*). Thus, the effect of pH on Cu isotope fractionation during the dissolution of Cu-bearing minerals can be discounted. In comparison, the influence of organic ligands seemingly existed in the organic matter-rich Cambisols. Organic ligands facilitate the release of isotopically heavy Cu into the soil solution through ligand-promoted dissolution, leaving behind $^{63}$Cu-enriched residual minerals (*Zheng et al., 2024*). Therefore, there will be a negative correlation between SOC content and soil $\delta^{65}$Cu values if the effect of organic ligands is the primary factor. Nevertheless, soil $\delta^{65}$Cu values demonstrated a positive correlation with SOC content across in all lands (Fig. 4C), suggesting that the role of organic ligands on soil $\delta^{65}$Cu composition was not dominant. In addition, the precipitation of Cu as carbonates is a prevailing process in calcareous soils (*Ponizovsky, Allen & Ackerman, 2007*), and may also occur in the Cambisols of this area. Secondary Cu-bearing carbonate minerals tend to be $^{63}$Cu-enriched and are prevail in the silt fraction (*Babcsányi et al., 2016*). However, a notable positive correlation between soil $\delta^{65}$Cu values and silt proportions in secondary forest land (Fig. 4B) implies that Cu precipitation as carbonate cannot explain the profile variations in soil $\delta^{65}$Cu composition.

The sorption of Cu(II)aq by metal (oxyhydr)oxides and clay minerals significantly influences Cu translocation and isotope fractionation during weathering and pedogenic processes (*Pokrovsky et al., 2008*). For the chemical weathering of carbonate rocks, although CIA values (59–76) being relatively low in soils (Fig. 2G), enrichment of Fe and Al and production of clay mineral were already occurring at this stage. Isotopically light Cu is preferentially sorbed by clay minerals (*e.g.*, kaolinite) and Mn oxyhydroxides (*e.g.*, birnessite) (*Ijichi, Ohno & Sakata, 2018*; *Li, Liu & Li, 2015*; *Sherman & Little, 2020*), while Fe (oxyhydr)oxides and gibbsite ($Al_2O_3.3H_2O$) are expected to sorb isotopically heavy Cu (*Pokrovsky et al., 2008*). The CIA values, $Fe_2O_3$ contents, and $Al_2O_3$ contents all exhibited negative correlations with soil $\delta^{65}$Cu values (Fig. 4), suggesting that the sorption of Fe (oxyhydr)oxides and gibbsite to isotopically heavy Cu was not dominant processes affecting soil $\delta^{65}$Cu composition. Soil $\delta^{65}$Cu values demonstrated negative correlations with MnO contents in the cropland and shrub-grass land, whereas positive correlations between them were observed in the secondary forest land and abandoned orchard (Fig. 4F). These results suggested that the sorption of Mn oxyhydroxides to isotopically light Cu significantly affected soil $\delta^{65}$Cu composition in cropland and shrub-grass land, but not in secondary forest land and abandoned orchard. Notably, clay fractions encompass both clay minerals and Fe, Al, and Mn (oxyhydr)oxides (*Babcsányi et al., 2016*). Given the negative correlation between soil $\delta^{65}$Cu values and clay proportions in the secondary forest land (Fig. 4A), it can be inferred that soil $\delta^{65}$Cu composition is significantly affected by the sorption of clay minerals to isotopically light Cu. Overall, Cu isotope fractionation through mineral sorption affects soil $\delta^{65}$Cu variation.

Organic complexation exhibits a stronger ability for Cu retention compared to the sorption of clay minerals (*Vialykh, Salahub & Achari, 2019*). Generally, isotopically heavy Cu(II) preferentially bonds with the carboxyl and carbonyl O ligands, while $^{63}$Cu is

enriched the Cu-SOM complexes formed through bonding with amino N ligands (*Ryan et al., 2014*; *Sherman & Little, 2020*). Plant fragments and polysaccharides, rich in carboxyl and carbonyl O ligands, are predominantly found in coarser (silt) fractions (*Quenea et al., 2009*). Silt fractions exceeded 75% (Fig. 2B), suggesting the dominance of carboxyl and carbonyl O ligands. This dominance resulted in a positive correlation between SOC content and soil $\delta^{65}$Cu values (Fig. 4C and Table 2). In comparison, amino N-containing compounds tend to accumulate and stabilize in clay fractions (*Mertz, Kleber & Jahn, 2005*), leading to an $^{63}$Cu-enrichment in organo-clay complexes. These $^{63}$Cu-enriched Cu-organo-clay complexes, along with clay minerals, migrate downward under leaching process, accounting for decrease in soil $\delta^{65}$Cu values with increasing soil depth (Fig. 3C). Moreover, the observed increase in CIA values with soil depth (Fig. 2G) is commonly associated with the downward translocation of secondary minerals in clay fractions due to leaching (*Mei et al., 2021*), confirming the prevalence of leaching in the study area. Additionally, abandoned cropland and orchard exhibited lower soil Cu contents and higher $\delta^{65}$Cu values compared to cropland (Fig. 3B), also attributed to the loss of $^{63}$Cu-enriched organo-clay complexes and/or clay minerals through leaching. Thus, Cu isotope fractionation, influenced by organic complexation, clay mineral sorption, and leaching process, primarily shape the $\delta^{65}$Cu patterns in soil profile (Fig. 5B).

In addition, Cu isotope fractionation in plant-soil systems can affect soil $\delta^{65}$Cu composition (*Zheng et al., 2024*). In natural lands, plants were significantly $^{63}$Cu-enriched compared to soils (Fig. 5B), mainly resulting from the preferential assimilation of isotopically light Cu in soils (*Ryan et al., 2013*). The $^{63}$Cu-enriched litter accumulates on the soil surface, subsequently leading to a decrease in the $\delta^{65}$Cu value in topsoil as litter decomposes (*Ren et al., 2022*). Conversely, there was a slight $^{65}$Cu-enrichment in topsoil (0–10 cm depth) compared to the deeper soils at the 10–20 cm depth in shrub-grass land and secondary forest land (Fig. 3C). This result is likely attributed to the minimal contribution of plants (which are not copper-accumulating vegetations) to soil Cu *via* litter return. Therefore, the $\delta^{65}$Cu signature derived from $^{63}$Cu-enriched plants was overprinted by Cu isotope fractionation in soils.

## $\delta^{65}$Cu patterns in different regions

This case study underscores the crucial roles of mineral sorption, organic complexation, and leaching process in affecting soil $\delta^{65}$Cu patterns. Each of these primary factors is intimately tied to climate conditions. To explore the $\delta^{65}$Cu patterns of Cambisols, correlations between their $\delta^{65}$Cu values and annual precipitation in different region (Table S1) were analyzed (Fig. 6). In humid climate regions, pedogenic processes predominantly involve the formation of secondary clay minerals, facilitated by abundant rainfall (*Mei et al., 2021*). This leads to the retention of isotopically light Cu through the sorption of clay minerals (*Li, Liu & Li, 2015*). Meanwhile, a moist soil environment accelerates the decomposition of $^{65}$Cu-complexed coarse organic matter (*Ryan et al., 2014*), leading to the loss of $^{65}$Cu(II)aq due to leaching. As a result, soil $\delta^{65}$Cu values declined with increasing annual precipitation in humid climates. In comparison, in subhumid climate regions, increased in rainfall may boost the formation of Fe (oxyhydr)oxides during pedogenic processes and the

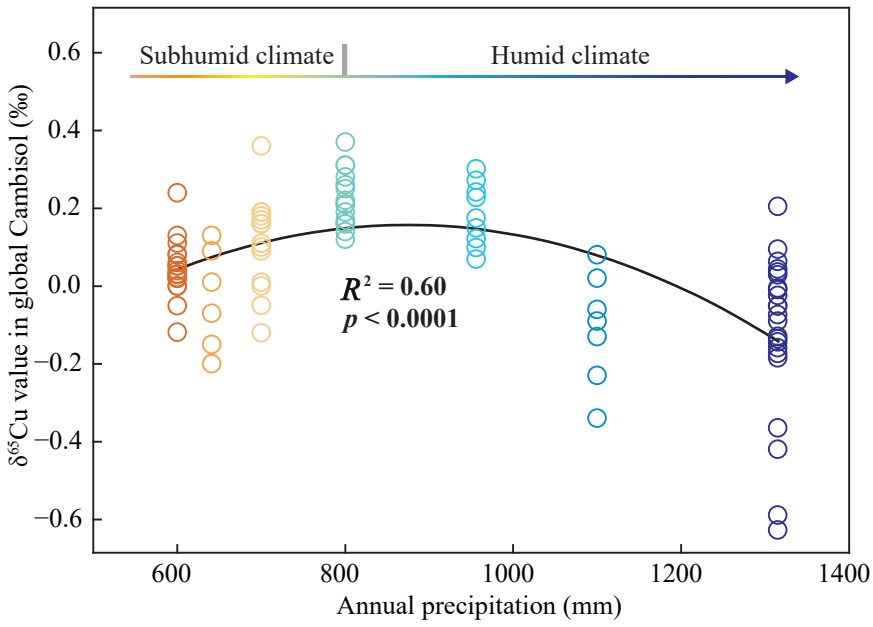

**Figure 6** **Relationships between the $\delta^{65}$Cu values of Cambisols and mean annual precipitation in different regions.** The data were derived from this study and other literatures (*Babcsányi et al., 2016*; *Bigalke et al., 2010*; *Bigalke, Weyer & Wilcke, 2011*; *Blotevogel et al., 2018*; *Fekiacova, Cornu & Pichat, 2015*; *Ren et al., 2022*).

accumulation of coarse organic matter as vegetation proliferates (*Merino et al., 2021*). Both processes favor $^{65}$Cu retention in soil through mineral sorption and organic complexation (*Li, Liu & Li, 2015*; *Ryan et al., 2014*). Therefore, soil $\delta^{65}$Cu values rise with increasing annual precipitation in subhumid climates. However, future research should delve into the relationships between soil $\delta^{65}$Cu values and other factors, such as bedrock types, soil properties, and vegetation types), to gain a more comprehensive understanding of Cambisol $\delta^{65}$Cu pattern and driving factors on a global scale.

## CONCLUSIONS

This study presented $\delta^{65}$Cu patterns in soil profiles and examined their correlations with soil Cu contents and physiochemical properties across different land uses in the karst region of Southwest China, focusing on the sources and biogeochemical processes of Cu in Cambisols. Findings suggested that soil Cu in agricultural lands predominantly originated from agricultural inputs (such as fungicides and farmyard manures) and bedrock, with a minor contribution from litter return. Soil $\delta^{65}$Cu values decreased with increasing soil depth, mainly attributed to Cu isotope fractionations through mineral sorption, organic complexation, and leaching process. In different regions, $\delta^{65}$Cu values in Cambisols showed a decreasing trend with rising annual precipitation in humid climate regions, while there was an inverse correlation between them observed in subhumid climate regions. This study

contributes to a fundamental understanding of Cu cycling in Cambisols under diverse land uses.

## ACKNOWLEDGEMENTS

We thank Rui Qu and Di Wang in China University of Geosciences (Beijing) and Wenxiang Zhou in The University of Queensland for the laboratory work.

### Funding

This study was supported by the "Deep-time Digital Earth" Science and Technology Leading Talents Team Funds for the Central Universities for the Frontiers Science Center for Deep-time Digital Earth, China University of Geosciences (Beijing) (No. 2652023001) and the National Natural Science Foundation of China (No. 41325010 & 42203011). The funders had no role in study design, data collection and analysis, decision to publish, or preparation of the manuscript.

### Grant Disclosures

The following grant information was disclosed by the authors:
"Deep-time Digital Earth" Science and Technology Leading Talents Team Funds for the Central Universities for the Frontiers Science Center for Deep-time Digital Earth, China University of Geosciences (Beijing): 2652023001.
National Natural Science Foundation of China: 41325010, 42203011.

### Competing Interests

The authors declare there are no competing interests.

### Author Contributions

- Man Liu conceived and designed the experiments, performed the experiments, analyzed the data, prepared figures and/or tables, authored or reviewed drafts of the article, and approved the final draft.
- Guilin Han conceived and designed the experiments, analyzed the data, authored or reviewed drafts of the article, and approved the final draft.
- Qian Zhang performed the experiments, authored or reviewed drafts of the article, and approved the final draft.

### Data Availability

   The raw measurements are available in the Supplementary File.

### Supplemental Information

Supplemental information for this article can be found online at http://dx.doi.org/10.7717/peerj.19982#supplemental-information.

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
