# Peer review of "Soil copper sources and biogeochemical processes under different land uses: insights from stable copper isotopes in the Cambisols of Southwest China"

_PeerJ, doi:10.7717/peerj.19982_

## Round 0.1 · original submission · Major Revisions

I recommend conducting a systematic literature search and evidence collection for the research area and land cover treated with Cu via fungicides. Add Cu accumulation and soil biota concerns to the introduction to improve its structure.

·

Basic reporting

The manuscript is well-written, professionally structured, and presented in clear scientific English. The background is well contextualized, references are adequate and up to date, and figures are informative. I suggest some minor revisions before acceptence:

1. Please ensure consistent notation of stable isotope values. The abstract and body of the text sometimes use “·65Cu” instead of “δ⁶⁵Cu” (e.g., Line 20). Use standard notation consistently.
2. Some in-text references lack author names (e.g., Lines 53, 57). Replace “et al.” with the first author’s last name when the authors are 3 or more.
3. Improve clarity of figure legends, particularly Figure 5 and Figure 6. More detailed explanations of axes, isotope endmembers, and mixing lines would enhance understanding for a broad audience.
4. Lines 185–193: Please briefly justify and reference the selection of Ti as the reference element for the enrichment factor (EF). While standard in geochemistry, this rationale should be stated explicitly in the methods. (e.g., Yuan et al., 2020 https://doi.org/10.1016/j.jhazmat.2020.122377)

5. Although the authors analyzed plant leaves or grasses, future research would benefit from sampling multiple plant compartments (roots, stems, etc.) to better constrain the plant contribution to soil Cu.
6. Lines 254–260: In some land uses (e.g., SG and AO), the range of δ⁶⁵Cu values is narrow, limiting statistical robustness. Please clarify this limitation and how it may influence the interpretation of regression analyses
7. The global comparison in Figure 6 is novel, but please clarify the dataset sources and control for differences in soil types and climate zones.

Experimental design

no commen

Validity of the findings

no commen

Additional comments

no commen

·

Basic reporting

Dear authors
I read with interest your manuscript entitled Soil copper sources and biogeochemical processes
under different land uses: Insights from stable copper isotopes in the Cambisols of Southwest China.

Your investigation is relevant, some points need your attention to be at the level of the journal and well-prepared to reach the scientific community.

I suggest conducting a brief systematic literature search and collating evidence at least for the study area and land cover subjected to Cu addition via fungicides. Improve the structure of the introduction by introducing the issue of Cu accumulation and the main issues for soil biota.

the reason for Cambisol or the other soil of the region must be explained (e.g because Cambisol are 80% of the soils in the study area) or dominant soil type. Use the WRB or soil taxonomy for more details.

I suggest adding a soil profile picture of the study site and its description. This will clarify many aspects for the specialised audience in soil science.

Global cambisol needs to be better explain, does it refer to world data?


Kind regards

Experimental design

Methods: is linear regression the best you can do with the data?
why not multiple linear regression and backward and stepwise to improve the model?

In the literature many advance techniques are applied I would like to stress that the right methodology can allow to discover patters and potential causal relationships, that at the moment we cannot see.

Validity of the findings

I suggest trying something more sophisticated in line with the literature analysis of similar studies

Additional comments

I will review the revised version

---

## Round 0.2 · accepted · Accept

The authors have addressed all of the reviewers' comments. This revised version is suitable for publication.

·

Basic reporting

The mauscript has been improved since the revisions, and no further revisions are needed before publication.

Experimental design

no comment

Validity of the findings

no comment

Additional comments

no comment